# Appropriate Use of Antibiotics in Acute Pancreatitis: A Scoping Review

**DOI:** 10.3390/antibiotics13090894

**Published:** 2024-09-18

**Authors:** Josep M. Badia, Sara Amador, Carmen González-Sánchez, Inés Rubio-Pérez, Alba Manuel-Vázquez, Montserrat Juvany, Estela Membrilla, José M. Balibrea, Xavier Guirao

**Affiliations:** 1Department of Surgery, Hospital General Granollers, School of Medicine, Universitat Internacional de Catalunya, Av Francesc Ribas 1, 08402 Granollers, Spain; samador@fphag.org; 2Department of Surgery, Hospital Universitario de Salamanca, 37007 Salamanca, Spain; crmngs@hotmail.com; 3Department of Surgery, Hospital Universitario La Paz, 28046 Madrid, Spain; dr.inesrubio@gmail.com; 4Department of Surgery, Hospital Universitario de Getafe, Getafe, 28905 Madrid, Spain; alba_manuel_vazquez@hotmail.com; 5Department of Surgery, Hospital del Mar, 08036 Barcelona, Spain; mjuvanygomez@gmail.com (M.J.); estelabe@gmail.com (E.M.); 6Department of Surgery, Hospital Universitari Germans Trias, 08916 Badalona, Spain; balibrea@gmail.com; 7Department of Surgery, Hospital Universitari Parc Taulí, 08208 Sabadell, Spain; xguirao@gmail.com

**Keywords:** acute pancreatitis, infected pancreatic necrosis, antibiotics, microbiology, prophylactic therapy, duration of treatment

## Abstract

Background: While selective use of antibiotics for infected pancreatic necrosis (IPN) in acute pancreatitis (AP) is recommended, studies indicate a high rate of inadequate treatment. Methods: A search of PubMed, Scopus, and Cochrane databases was conducted, focusing on primary research and meta-analyses. Data were categorized based on core concepts, and a narrative synthesis was performed. Results: The search identified a total of 1016 publications. After evaluating 203 full texts and additional sources from the grey literature, 80 studies were included in the review. The answers obtained were: (1) Preventive treatment does not decrease the incidence of IPN or mortality. Given the risks of bacterial resistance and fungal infections, antibiotics should be reserved for highly suspected or confirmed IPN; (2) The diagnosis of IPN does not always require microbiological samples, as clinical suspicion or computed tomography signs can suffice. Early diagnosis and treatment may be improved by using biomarkers such as procalcitonin and novel microbiological methods; (3) When indicated, early initiation of antibiotics is a key determinant in reducing mortality associated with IPN; (4) Antibiotics with good penetration into pancreatic tissue covering Gram-negative and Gram-positive bacteria should be used. Routine antifungal therapy is not recommended; (5) The step-up approach, including antibiotics, is the standard for IPN management; (6) Antibiotic duration should be kept to a minimum and should be based on the quality of source control and patient condition. Conclusions: Early antibiotic therapy is essential for the treatment of IPN, but prophylactic antibiotics are not recommended in AP. High-quality randomized controlled trials are required to better understand the role of antibiotics and antifungals in AP management.

## 1. Introduction

Acute pancreatitis (AP) is a life-threatening condition with an inflammatory onset that presents with a wide range of severity, from mild to moderate or severe disease [1]. Severe cases are closely linked to the presence of necrosis and may lead to organ failure, infection, and the development of infected pancreatic necrosis (IPN). The Atlanta classification Criteria (revised in 2012) standardised the diagnosis and classification of AP based on clinical symptoms, serum amylase or lipase levels, and imaging findings [2].

Bacterial translocation, in which bacteria migrate from the intestine to the pancreas, contributes to the systemic inflammatory response in AP and is considered the cause of infection of necrotic material [3]. In recent decades, the use of antibiotics to prevent or treat IPN has become a controversial topic. Although early randomised studies and meta-analyses suggested benefits of antibiotic prophylaxis, subsequent trials and systematic reviews have not confirmed these findings. Currently, most clinical guidelines only recommend antibiotics when infection is confirmed or strongly suspected. Despite these recommendations, recent studies have found a high rate of inappropriate antibiotic use in AP [4,5].

This scoping review aims to update the available scientific evidence on the use of antibiotics in the prophylaxis and treatment of AP. It assesses IPN microbiology, diagnostic criteria, antibiotic/antimicrobial selection, and duration of treatment.

## 2. Results

Figure 1 shows the PRISMA flowchart of the review. In total, 1016 publications were identified. After removing duplicates and screening by title/abstract, 813 publications were excluded and 203 were selected for full-text screening. Of these, 133 were subsequently excluded after retrieval, leaving 70 studies for evaluation. Furthermore, ten additional publications from sources indicated in the methodology were incorporated into the analysis. After adding these studies, 80 articles were analysed, of which 13 were systematic reviews or meta-analyses and 22 were randomised controlled studies (RCTs). Figure 2 shows the data map of the key issues reviewed.

### 2.1. Definitions, Classification and Diagnosis

AP is an acute inflammatory process of the pancreas with a mortality ranging from 1% in oedematous interstitial AP to up to 28% in severe AP [6,7,8]. According to the 2012 Atlanta classification [2], acute necrotising pancreatitis is characterised by inflammation associated with pancreatic parenchymal necrosis and/or peripancreatic necrosis. Between 5–10% of patients develop necrosis of the pancreatic parenchyma, peripancreatic tissue, or both. Necrotising pancreatitis can usually only be detected 72 to 96 h after the onset of symptoms [9], and is associated with high rates of early organ failure (38%), need for intervention (38%) and death (15%) [10].

### 2.2. Prediction of Severity

Various grading systems are available to predict the severity of AP with little clinical impact. Most clinical guidelines recommend strict monitoring of systemic inflammatory response syndrome (SIRS) and organ failure parameters during the first 48 h (especially renal, respiratory and cardiovascular failure) [2,10,11], using the Marshall [12] or SOFA [13] scales.

The Atlanta criteria uses the existence of organ failure to classify severity, categorising AP as mild (no organ failure or local/systemic complications), moderate (no organ failure or transient organ failure <48 h and/or local complications) or severe (persistent organ failure >48 h that may involve one or more organs).

### 2.3. Local Complications

Local complications should be suspected when there is persistent pain, elevated lipase or amylase, persistent organ failure, fever fever and leukocytosis [2]. Pancreatic and peripancreatic necrosis can remain sterile or become infected, with no absolute correlation between the extent of necrosis and the risk of infection [2,8]. Necrosis presents an infection rate of 33% and mortality ranges from 15% to 35% [14]. IPN is rare during the first week, when sepsis is mainly due to pneumonia or bacteraemia [8]. There is no correlation between the extent of necrosis and the risk of infection. Although infection can occur early in the course of necrotising pancreatitis, it is most often seen late in the clinical course, after 10 days, and peaks between the second and fourth week of AP [8,15,16].

### 2.4. Infections and Microbiology of AP

The mean time from onset of AP to diagnosis of IPN in the PROPATRIA study was 26 days [17]. However, other authors using routine fine needle aspiration (FNA) detect it in most cases at two weeks, so it is possible that IPN may develop before the clinical manifestation of infection [18,19]. In addition to IPN, up to 20% of patients with AP develop extrapancreatic infections (bacteraemia, sepsis, pneumonia or urinary tract infections) [8], which are associated with increased mortality [20].

In previously bacteraemic patients with necrosis, the risk of IPN is increased and in about half of these patients, the organism isolated from both sources is the same [6]. Bacteraemia is an independent mortality and risk factor for IPN and can be used as a prognostic factor in patients with known pancreatic necrosis [6].

Most infections (approximately 75%) are monomicrobial, caused by organisms of enteric origin [3]. The main source of infection is thought to be bacterial translocation from the small intestine rather than from the colon [21]. At the time of initial infection, Gram-negative bacteria (especially *Escherichia coli*, *Klebsiella* spp. and *Pseudomonas aeruginosa*), Gram-positive bacteria (*Enterococcus faecium*, *Enterococcus faecalis*, *Staphylococcus epidermidis, Staphylococcus aureus*) and fungi (*Candida* spp.), with some regional variation [8,22,23], should be considered. Initial fungal infection can double the mortality rate [8].

The presence of multi-drug resistant organisms (MDROs) is of concern in this patient group, especially with prolonged empirical treatments. MDRO rates of 21–32% have been reported in necrotic tissue from patients treated prophylactically with a carbapenem [24,25]. In a 2018 study, 50% of patients acquired an extensively drug-resistant bacterial infection at some point, which was the main reason for prolonged ICU stay [26].

*Gut microbiota dysbiosis in AP.* Manipulation of the gut microbiome may be a new line of treatment, as a dysbiosis of the gut microbiota in AP has been identified [27,28,29], based on an apparent reduction in bacterial diversity in this site [29]. The gut microbiome may be altered in AP patients, characterised by a decrease in Bifidobacterium and bacteria that produce short-chain fatty acids with anti-inflammatory properties and, in contrast, an increase in harmful bacteria such as Enterobacteriaceae and *Enterococcus* [28,29].

### 2.5. Management of IPN. Research Questions

Table 1 summarises the main findings of the scoping review.

#### 2.5.1. Does Preventive Antibiotic Treatment Reduce Mortality in AP with Necrosis?

The possibility of preventing IPN and its associated morbidity and mortality by prophylactic administration of antibiotics has been a matter of debate for 40 years. This preventive antibiotic treatment (PAT) has been the subject of at least 21 RCTs and 21 meta-analyses and is a topic of discussion in all AP clinical practice guidelines. Although early RCTs and meta-analyses showed reductions in mortality, incidence of IPN and extrapancreatic infections, subsequent higher-quality studies and larger numbers of cases have changed the paradigm. The most recent guidelines [1,9,30,31,36,37,38,39] do not recommend PAT in AP of any degree of severity, nor when pancreatic necrosis is found; this is because the results of meta-analyses published since 2008 found no difference in mortality or incidence of IPN with the use of antibiotics [40,41,42,43,44,45,46].

As mentioned, an overall review of the published meta-analyses on PAT shows no reduction in the incidence of necrotising infection or mortality, although most of these studies showed a fall in extrapancreatic infections [41,42,44,45,46,52,53,54], especially due to a decrease in sepsis and urinary tract infections. However, a rigorous assessment of the results of the published meta-analyses should take two factors into account: the methodological quality of the RCTs on which they are based, and the sample size. De Vries et al. analysed the quality of RCTs reporting on the effect of PAT in AP prior to 2009 and found an inverse relationship between the methodological quality of the RCTs and the reduction in mortality risk, categorising their overall quality as moderate [55]. To address the sample size factor, an important reference point is the systematic review and meta-analysis by Poropat et al. published in 2022 [46], which demonstrated that current meta-analyses of mortality, sepsis and urinary tract infection outcomes are underpowered, and as a result are unable to provide strong evidence of the impact of PAT on these conditions. The only positive results with the use of PAT were reductions in hospital stay, overall infection rates and extrapancreatic infections, findings that were in agreement with most previous meta-analyses.

In summary, there is no evidence that PAT decreases the main outcomes of infection prevention and mortality in pancreatic necrosis, but we cannot be sure that this lack of evidence is not due to the possibility that the sample of available RCTs and meta-analyses is insufficient. Nevertheless, the authors believe that the recommendations of current guidelines should be followed and that signs of severity or necrosis are not sufficient grounds for routine TAP. However, subtle warning signs that may raise suspicion of IPN should be recognised early and antibiotic treatment should be initiated in these situations.

Further high-quality studies and adequate sample sizes are therefore needed to reach the targets calculated by Poropat et al. of 1638 cases to analyse mortality, 1291 to analyse sepsis and 871 to analyse urinary tract infection [46].

#### 2.5.2. How Is IPN Diagnosed?

For suspected diagnosis, a study of the value of routine laboratory tests to differentiate infected and sterile necrosis found only C-reactive protein (CRP) and leukocyte counts to be discriminatory, with cut-off values of 81 mg/L for CRP and 13 × 109/L for leukocytes [56]. However, in the early phase of AP, the elevation of these parameters is indistinguishable from that caused by other infectious complications or the inflammatory state of the disease itself [1,57,58] and is not considered sufficient to diagnose IPN and thus indicate antibiotic treatment.

IPN may be suspected in patients with pancreatic or extrapancreatic necrosis with clinical decline (clinical instability, persistent sepsis, new or persistent organ failure, increased need for intensive support, leucocytosis, rising CRP or fever) or who do not improve after 7–10 days of hospitalisation. In this context and in the absence of an alternative cause of infection, a retrospective study showed that IPN caused the condition in 80% of cases, with a false positive rate of 20% [32].

In 2014 systematic review considered procalcitonin (PCT) to be the best biological predictor of IPN, with a cut-off value of 3.5 ng/mL offering a sensitivity of 90% and a specificity of 89%, although its values appear to be slightly higher than the upper normal limit [34]. Most of the evidence in favour of PCT does not come from RCTs; a 2017 Cochrane study could not reach conclusions on the usefulness of CRP and PCT due to the paucity of RCTs and the methodological shortcomings of those available [59]. Subsequently, an RCT analysed a PCT-based algorithm for deciding the initiation, continuation and termination of antibiotic treatment, finding a decrease in antibiotic use in the PCT-guided group, and no increase in the rate of infections and mortality [60]. Recent guidelines consider that PCT may be the most useful marker for predicting the risk of developing IPN [1,10,30,35].

When IPN is suspected, the first examination performed is a CT scan, in search of extraluminal gas in the pancreatic or peripancreatic tissues. The presence of gas in these collections suggests a direct diagnosis of infection with a very high degree of certainty; however, gas formations occur in only half of patients with IPN (sensitivity 56%, specificity 97%) [1,2,19,61].

Diffusion-weighted MRI also visualises air in the retroperitoneum [62], but it is not always available and is difficult to perform in critically ill patients. Furthermore, few studies are available on its superiority over CT [10,61].

There is no doubt that the diagnosis can be established with the detection of bacteria or fungi by Gram stain or culture on fluid obtained by FNA, although it presents false negative rates of 20–25% and false positive rates of 4–15% [32,63]. Furthermore, FNA may be associated with iatrogenic infectious complications such as peritoneal contamination and gastrointestinal perforation and haemorrhagic complications. It is therefore no longer routinely performed at many centres [1] and current guidelines do not consider it essential for the diagnosis of IPN [1,9,10,30].

However, new methodologies ancillary to microbiology, such as metagenomic next-generation sequencing (mNGS), may revolutionise the management of IPN by facilitating earlier diagnosis, as well as broadening the range of pathogens identifiable in the sample. In a retrospective study of 40 patients with suspected IPN undergoing CT-guided FNA, culture and mNGS were used simultaneously. The mNGS result was obtained earlier than the culture result (42 h (36–62 h) vs. 60 h (42–124 h), P = 0.032); furthermore, traditional cultures isolated seven bacterial species and two fungal species, while mNGS detected 22 bacterial species and two fungal species. The sensitivity, specificity, negative predictive value (NPV) and positive predictive value (PPV) of mNGS were 88.0%, 100%, 83.3% and 100%, respectively [64]. More recently, other authors hypothesised that the addition of mNGS to the standard FNA procedure may improve diagnostic accuracy. In a prospective cohort of 27 patients, the mNGS technique had an advantage in timeliness, but no significant difference was found between mNGS and culture approaches in the positive rate. Although mNGS results led to a change in treatment in 80% of patients, no clinical benefit was observed compared with historical controls [65].

In a recent retrospective study, 58% of mNGS tests were found to be highly useful, leading directly to changes in antimicrobial therapy, selection of therapy duration, targeting of new diagnoses or avoidance of further diagnostic needs [66].

This same mNGS technique has even been used to analyse circulating microbial cell-free DNA. In plasma from 44 patients with suspected IPN, the mNGS positivity rate was 54.6%. In patients in whom IPN was confirmed, mNGS was compared with microbiological results, which were considered the reference standard. Of 24 cases with positive mNGS, 20 (83.3%, 95%CI 68.42–98.2%) matched the results with the IPN drainage culture. The positive and negative percentage agreements of plasma mNGS for IPN were 80.0% (95%CI 64.32–95.68) and 89.5% (95%CI 75.67–100), respectively. Furthermore, compared with the mNGS-negative group, patients in the positive group had more new-onset septic shock [33]. Further experience is probably needed before this technology can be integrated into practice.

#### 2.5.3. When Should Antibiotic Treatment Be Prescribed in AP?

When infection is suspected, antibiotics should be started promptly while the source of infection is being determined. Using data from RCTs prior to 2015, two meta-analyses suggest that, in the presence of IPN, early initiation of treatment is a key determinant in reducing mortality [47,53]. As described above, certain clinical signs and the presence of retroperitoneal gas are reasonably suggestive of IPN and empirical antibiotic treatment can be initiated without FNA, especially if percutaneous drainage is to be part of the management algorithm [1,9,31,48]. However, if cultures are subsequently negative and no source of infection is identified, antibiotics should be discontinued.

#### 2.5.4. Which Antibiotics Are Indicated in the Treatment of IPN?

The choice of antibiotic should be based on the results of in vitro antimicrobial sensitivity and in vivo bioavailability, including penetration into pancreatic tissue. Pancreatic necrosis, in which vascular supply is lacking, would be expected to have minimal penetration. However, some studies have detected significant levels of piperacillin/tazobactam, metronidazole, imipenem, ciprofloxacin and ofloxacin in necrotic pancreatic tissue samples [49,50,67]. It should be remembered that these broad-spectrum molecules may be responsible for the emergence of MDRO and that there is little literature analysing other antibiotics with less ecological impact [30].

If empirical antibiotics are initiated, agents with Gram-negative and Gram-positive coverage should be used (e.g., a carbapenem alone; or piperacillin/tazobactam, ceftazidime or cefepime combined with an anaerobic agent such as metronidazole). When microbiology results are available, antibiotic therapy should be tailored according to the pathogens identified and their susceptibility patterns. This personalised approach helps to optimise therapeutic efficacy and minimise the development of antibiotic resistance.

*Antifungals.* Antifungal prophylaxis has been suggested for patients receiving broad-spectrum antibiotics. The use of prophylactic antibacterial therapy, and especially its long duration, increase the incidence of pancreatic fungal infection. A 2021 meta-analysis reported a 26.6% incidence of fungal infections in IPN [68], with Candida albicans being the most frequently isolated fungus, but it should be noted that many of the included patients had prolonged prior antibiotic therapy.

It seems clear that fungal infection negatively affects the prognosis of patients with pancreatitis and is associated with increased morbidity, mortality, ICU admission rate and length of hospital stay. However, antifungal agents may not reach therapeutic levels in poorly perfused pancreatic or peripancreatic tissues, and current guidelines do not recommend the routine administration of prophylactic antifungal therapy in conjunction with therapeutic antibiotics [1,10,30,31,39,69].

#### 2.5.5. What Is the Adequate Duration of Antibiotic Treatment?

There are no data on the optimal duration of antibiotic treatment in IPN. Although early and aggressive antibiotic therapy is essential, its prolonged use should be justified based on the clinical response and ongoing infection. Duration of antibiotic treatment can be guided by factors such as extent of necrosis, clinical improvement and the resolution of systemic inflammatory markers. Although still a matter of debate, it is common to discontinue antimicrobials 48 h after removal of the last drain if all cultures are negative [10]. Individualised patient assessment, taking into account factors such as immunocompetence and comorbidities, is crucial in determining the appropriate duration of antibiotic treatment.

#### 2.5.6. What Is the Role of the Step-Up Approach?

The step-up approach has been proposed as an effective strategy in the treatment of IPN. It is based on a combination of therapies, starting with conservative measures, such as antibiotics and supportive management, and gradually escalating to more invasive interventions, such as ultrasound- or CT-directed percutaneous drainage, video-assisted retroperitoneal necrosectomy (VARD), transgastric necrosectomy or open surgery, if necessary.

The pioneering PANTER (Percutaneous Step-Up Approach in Necrotising Pancreatitis) trial showed that image-guided percutaneous drainage reduces the rate of serious complications or death when compared to open surgery for IPN [48]. Upon suspicion or confirmation of IPN, the step-up approach starts with antibiotic treatment. The next step is the insertion of an image-directed pig-tail drain through the left retroperitoneum, facilitating minimally invasive retroperitoneal necrosectomy, if necessary. If there is no clinical improvement, the third step, left retroperitoneal surgical step-up approach (LRRSA), is performed with continuous postoperative high-volume lavage.

In 2018, the TENSION trial showed that the endoscopic step-up approach is not superior to the surgical approach in terms of mortality or major complications, although the pancreatic fistula rate and hospital stay were lower with endoscopy [70]. The authors concluded that the endoscopic step-up approach is the treatment of choice for transgastric approachable necrosis and that the left retroperitoneal surgical step-up approach should be reserved for IPN far from the stomach, alone or in combination with the endoscopic approach. Whether performed endoscopically or surgically, it is best to delay the step-up approach as long as possible for optimal results [71].

## 3. Discussion

This paper provides a comprehensive overview of the existing literature on the role of antibiotics in the management of AP. It summarises the available evidence related to risk profiles, diagnosis of IPN, microbiology of infection, the most appropriate antibiotics and criteria for their prescription, operative and non-operative management, and duration of treatment. The paper demonstrates the need for further high-level studies assessing most aspects of practice.

While the main finding of the review is that the presence of SIRS criteria and organ failure predicts severity, AP mortality is strongly correlated with extrapancreatic infections and IPN. These complications may occur at any time during the course of the disease but are most frequent after 8–10 days of admission, although IPN probably appears earlier than is suggested in current reports. Obtaining microbiological samples by FNA is not always essential for its diagnosis, and a high level of clinical suspicion or CT signs may be sufficient to initiate antibiotic therapy. The diagnostic suspicion of IPN in patients with pancreatic necrosis could be based on further clinical decline with worsening organ failure, accompanied by increased CRP and leukocyte levels, while the most useful biomarker in this context seems to be PCT. Promising microbiological methods such as metagenomic sequencing are currently emerging that may allow a prompt diagnosis of IPN and the establishment of early antibiotic therapy, which has been correlated with therapeutic efficacy. It is possible that mNGS-based methods have higher positive rates than conventional culture methods, and the possibility of their implementation in plasma opens up a relevant field of clinical experimentation, reinforcing the message that FNA can be avoided. At present, however, the interpretation of mNGS data needs to be combined with clinical data and conventional methods. Emerging research on intestinal dysbiosis in AP, the possible manipulation of the intestinal microbiome and the effect of selective decontamination of the intestinal tract raise novel issues in the antibiotic management of AP that also deserve further investigation.

Based on updated information from meta-analyses, all current clinical guidelines recommend antibiotic therapy only in cases of highly suspected or confirmed IPN. This recommendation seems to be supported by the present review, although the issue remains open for further research and cannot be considered settled. Thus, although the currently available meta-analyses do not present evidence that PAT decreases pancreatic necrosis infection and mortality, this possibility cannot be ruled out due to the insufficient sample size in the published RCTs and the low statistical power of meta-analyses assessing mortality and infections of all types in PC. As discussed, the conclusions of the Poropat et al. [46] meta-analysis are illustrative, as they used the trial sequential analysis technique to calculate the sample size necessary for the results of a meta-analysis to be considered reliable. This technique and the estimation of the optimal sample size is described in detail in García-Alamino et al. [72]. Calculating the optimal sample size reduces the risk of false results in meta-analyses and determines whether more RCTs are needed to address the effects of the intervention or, conversely, whether a given meta-analysis can be considered as providing definitive evidence. Poropat et al. believed that no distinction can be made between the lack of efficacy of PAT and the lack of power of meta-analyses, a circumstance that may lead to the erroneous rejection of an effective intervention because of a type 2 statistical error (i.e., a false negative result). This may be one of these situations in which absence of evidence is not evidence of absence. Some beneficial effects have, however, been demonstrated with PAT, such as reductions in hospital stay and in rates of general and extra-pancreatic infections.

Despite the theoretical agreement between meta-analyses and guidelines in terms of knowledge of scientific evidence, there is a notable gap with regard to the practice of care. A British study reported PAT in 62% of AP without a clear explanation of its indication [5]. Another retrospective study, conducted in 23 countries, showed an overall overuse of antibiotics in AP ranging from 31% to 82%, with higher rates in Asia [35]. No consensus was detected for the indication of antibiotic therapy, initiated based on leukocytosis, lipase or amylase and CRP. The same study found that these parameters were always elevated in the early stages of pancreatitis, but were not related to infection, unlike PCT, which was a good predictor of infection at this stage.

With regards to IPN source control, this issue has been studied extensively in the last decade, over the course of which the concept of the step-up approach has been consolidated. Antibiotic treatment with Gram-positive, Gram-negative and anaerobic coverage should be considered as the first stage of the step-up approach, gradually followed by image-guided percutaneous drainage and retroperitoneal or transgastric necrosectomy, which should be postponed as long as the patient’s condition allows.

As in all intra-abdominal infections, the duration of antibiotic treatment should be as short as possible, depending on the quality of source control and the patient’s clinical condition. Although outside the time period of this review, recent Surgical Infection Society guidelines recommend limiting therapy to four days for low- and high-risk patients even when source control has been achieved by a percutaneous drainage procedure [51].

Our review has some limitations. We only included studies written in English or Spanish. Studies that reported qualitative data that might have been relevant to the findings were excluded. Other relevant limitations, which need to be addressed in future research, include the heterogeneity of the studies in terms of differences in protocols, outcomes, and small sample sizes of the trials. Finally, due to the use of the scoping review methodology [73], no quality assessment of the studies included was performed.

## 4. Materials and Methods

This study is based on the guidance framework for conducting scoping reviews developed by the Joanna Briggs Institute [74], and is reported in accordance with the Preferred Reporting Items for Systematic Review and Meta-Analysis (PRISMA) extension for scoping reviews (PRISMA-ScR) [73]. The project was preregistered on the OSF Database (https://osf.io/fsk7a/?view_only=, accessed on 29 July 2024). PubMed, Scopus and Cochrane databases were searched using a predefined strategy. The reference lists of all included studies were manually reviewed to identify other relevant papers. The records retrieved were double-screened for eligibility.

### 4.1. Eligibility Criteria

Inclusion criteria were studies of patients with AP and interventions such as antibiotic prophylaxis, and the treatment and management of pancreatic infection. The studies included were systematic reviews with or without meta-analysis and RCTs. Narrative reviews, case reports, observational case-control studies, editorials, expert opinions, pre-clinical studies, conference proceedings and studies not published in English or in Spanish were excluded.

### 4.2. Search Strategy

The studies were searched using relevant keywords and Medical Subject Heading (MeSH) terms spanning the period from January 2000 to December 2023. Additional references were identified through manual searches from the grey literature, analysing key articles and reference lists of relevant studies; searches of guidelines of professional associations and societies; and Google Scholar, ResearchGate and Academia.edu for reports and preprints.

### 4.3. Selection of Sources of Evidence

Screening was carried out independently by two reviewers (JMB and SA), with full texts of potentially relevant articles considered for inclusion. Disagreements between reviewers were resolved by consensus or with a third reviewer (MJ). Study descriptors were extracted, including title, first author, year of publication, country of origin, study design, sample size, study setting (single or multicentre) and key findings. In order to group the papers identified, key themes from the studies were identified and used to create a framework corresponding to the clinical pathway, addressing aspects such as diagnoses, treatments and outcomes.

### 4.4. Synthesis of Results

A narrative synthesis of the studies included was produced. In line with the methodology of scoping reviews, a formal assessment of the risk of bias in the studies was not performed [73]. Identification of gaps was undertaken by reviewing areas addressed by the studies according to the aspect of care addressed. Study findings were considered by the research team, and questions relevant to the gaps in the evidence were proposed. A concept map was generated with a data mapping tool to visualize the evidence and demonstrate any gaps in the evidence (MapForce^®^ 2024).

### 4.5. Research Questions

A summary of the published evidence on definitions, severity criteria, complications and microbiology of pancreatic infection was created. In addition, a number of research questions were formulated: (1) Does antibiotic prophylaxis reduce mortality in AP with necrosis? (2) How should be IPN diagnosed? (3) When should antibiotic treatment be prescribed in AP? (4) Which antibiotics are indicated in the treatment of IPN? (5) What is the adequate duration of antibiotic treatment? (6) What is the role of the step-up approach?

### 4.6. Endpoints

The following endpoints were used in the analysis of the selected articles: mortality rate; pancreatic necrosis infection rate; non-pancreatic infection rate; sepsis; need for surgical intervention; and length of hospital stay (LOS).

## 5. Conclusions, Data Gaps and Implications for Research and Practice

Current evidence does not favour the routine use of preventive antibiotics in severe AP or AP with necrosis. Guidelines suggest treatment only when IPN is confirmed or strongly suspected. The results of available RCTs and meta-analyses should be considered inconclusive due to the heterogeneity of the studies, with variations in antibiotic classes, timing of treatment, outcomes analysed, study designs and, above all, sample sizes. As the early initiation of antibiotic therapy is of crucial importance in the context of IPN, the question remains whether certain subgroups of patients (e.g., those with severe necrotising pancreatitis) could benefit from early antibiotic use, particularly in high-risk groups. Our understanding of the use of antibiotics in AP seems incomplete; evidence is lacking on the overall efficacy of prophylactic or preventive antibiotics in IPN, the criteria for initiation of treatment, the type of antibiotics to be used and their timing. Furthermore, robust evidence comparing different antibiotic regimens is lacking, particularly for Gram-negative and anaerobic organisms. There is also limited data on the pharmacokinetics and tissue penetration of antibiotics in the pancreas, particularly in necrotising pancreatitis. Assessment of PCT and CRP levels and some novel microbiological techniques such as the use of mNGS in plasma may be useful to predict the likelihood of developing IPN and to identify patients who might benefit from early antibiotic treatment.

Future research should aim to fill these gaps through well-designed studies focused on patient subgroups, the impact of antibiotics on the microbiome, antimicrobial resistance and the long-term consequences of antimicrobial use. With growing concerns about antibiotic resistance, it is crucial to refine antibiotic practices in AP to ensure effective treatment while minimising unnecessary exposure. Although it is advisable to adhere to the current guidelines for sparing antibiotic use in AP, high-quality RCTs are now needed to shed light on these unclarified issues regarding the role of antibiotics and antifungals in AP.

## Figures and Tables

**Figure 1 antibiotics-13-00894-f001:**
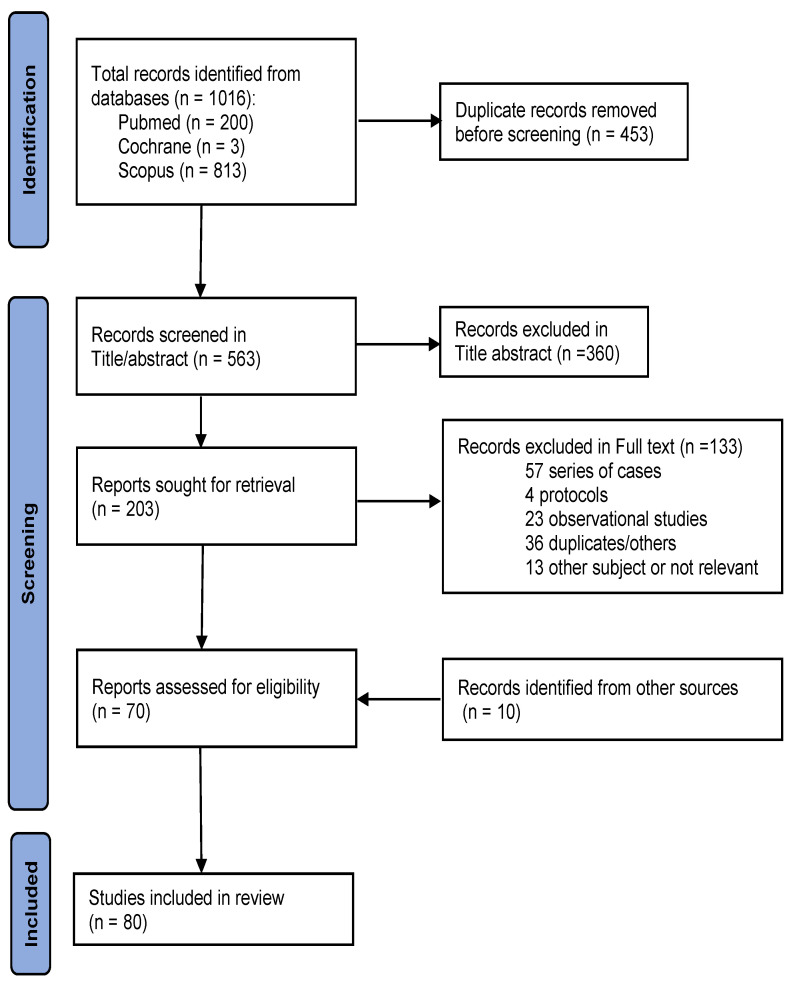
Preferred Reporting Items for Systematic Reviews and Meta-Analyses (PRISMA) flow diagram for the scoping review process.

**Figure 2 antibiotics-13-00894-f002:**
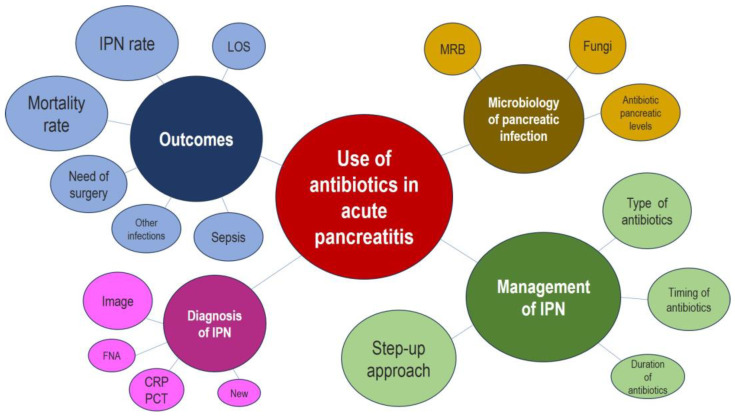
Data mapping to key antibiotic use in acute pancreatitis themes. IPN: infected pancreatic necrosis; LOS: length of hospital stay; FNA: fine needle aspiration; CRP: C-reactive protein; PCT: procalcitonin; MRB: multi-resistant bacteria.

**Table 1 antibiotics-13-00894-t001:** Summary of the main results of the scoping review.

Findings	References
Prediction of severity should be based on the existence of SIRS and organ failure criteria.	[2,9,10]
Obtaining microbiological samples by FNA is not essential for IPN diagnosis.	[1,9,10,30]
Clinical suspicion and CT signs may be sufficient to initiate antibiotic therapy for IPN.	[1,9,10,30,31,32,33]
The most useful biomarker for IPN diagnosis seems to be procalcitonin.	[1,10,30,34,35]
Most infections are monomicrobial, caused by organisms of enteric origin.	[3,8,22,23]
Routine prophylactic antibiotics should be not prescribed for patients with necrotizing acute pancreatitis.	[1,9,30,31,36,37,38,39,40,41,42,43,44,45,46]
Infection with multidrug-resistant organisms or fungi may follow prophylactic antibiotic therapy.	[8,24,25,46]
Antibiotic therapy should be administered only to treat highly suspected or confirmed IPN.	[1,9,30,31,36,37,38,39]
Agents with Gram-negative and Gram-positive coverage should be used for IPN therapy.	[30,47,48,49]
Routine prophylactic antifungal therapy in combination with therapeutic antibiotics is not recommended.	[1,9,10,30,31,39,50]
Duration of antibiotic treatment should be as short as possible, depending on the quality of septic source control and the patient’s clinical condition.	[10,51]

SIRS: systemic inflammatory response syndrome; FNA: fine needle aspiration; IPN: infected pancreatic necrosis; CT: computed tomography.

## Data Availability

All data will be made available on request.

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
