# Peer review of "Appropriate Use of Antibiotics in Acute Pancreatitis: A Scoping Review"

_antibiotics, 2024, doi:10.3390/antibiotics13090894_

Round 1
Reviewer 1 Report
Comments and Suggestions for Authors
This paper provides an overview of the existing literature on the role of antibiotics in the management of AP. Well summarized.
And I pay special attention to the description about mNGS application in AP. As a matter of fact, there are several other paper which has not been cited in this review, which provided evidence that mNGS might have the potential to be better than PCT in guiding antibiotics usage in acute necrotizing pancreatitis.
Comments on the Quality of English LanguageGood quality of English language
Author Response
This paper provides an overview of the existing literature on the role of antibiotics in the management of AP. Well summarized.
And I pay special attention to the description about mNGS application in AP. As a matter of fact, there are several other paper which has not been cited in this review, which provided evidence that mNGS might have the potential to be better than PCT in guiding antibiotics usage in acute necrotizing pancreatitis.
Thank you very much for your evaluation. We also believe in the future of the mNGS technique in the diagnosis of intfected necrosis. Following your suggestion, we added two more references on the subject and we extended the comments in discussion. We agree that it is a subject for further research and that it may be superior to PCT-based diagnosis.
Reviewer 2 Report
Comments and Suggestions for Authors
This scoping review covers an important topic, is well-written, structured, but would benefit from a clearer focus, particularly on key issues like clinical guidelines for antibiotic use in acute pancreatitis.
(1) The inclusion criteria are well-defined, though explaining exclusions would improve transparency. Strengthening the discussion on how the findings influence clinical practice and adding specific recommendations would enhance the paper’s impact.
(2) Expanding on data gaps and future research areas would provide more depth.
Comments on the Quality of English LanguageThe quality of English within the manuscript is fine, just few areas that need to be revised and the manuscript proofread.
Author Response
Comments 1. The inclusion criteria are well-defined, though explaining exclusions would improve transparency. Strengthening the discussion on how the findings influence clinical practice and adding specific recommendations would enhance the paper’s impact.
Thank you very much for your evaluation. The exclusion criteria that were used for the review are defined between lines 406 and 408 and the truth is that we have not been able to expand on this information, as all the criteria used in the study are included.
Comments 2. Expanding on data gaps and future research areas would provide more depth.
Following their suggestions, we have further elaborated the discussion on data gaps and future research areas in the Discussion and Conclusions.
Reviewer 3 Report
Comments and Suggestions for Authors
Nice work, enjoyed reading the manuscript. Hope my comments will be well received and easy to follow. Comments are mostly minor, however, it will improve the quality of the manuscript and will be easy for readers to follow. Suggestions are mainly clarifying few sections, terminology changes, etc.
Good luck

Manuscript is well written, easy to follow, and English Language quality is fine. However, I have suggested few minor editing and terminology changes for the authors to consider.
Author Response
Comment 1. Nice work, enjoyed reading the manuscript. Hope my comments will be well received and easy to follow. Comments are mostly minor, however, it will improve the quality of the manuscript and will be easy for readers to follow. Suggestions are mainly clarifying few sections, terminology changes, etc.
Dear reviewer, thank you very much for your comments and suggestions. Although the text had been reviewed by a British medical writter, it clearly needed further revision.
I must say that your review was, without doubt, one of the most comprehensive we have ever received, and was very much appreciated by us. It is indeed an excellent example for our future actions as peer reviewers.
We consider that the manuscript now has enhanced quality and clarity, and hope that all the modifications we have made will meet with your satisfaction.We thank you again for taking the time to review our article so thoroughly.
